# A Theoretically Grounded Extension of Universal Attacks from the Attacker's Viewpoint

## Abstract

We extend universal attacks by jointly learning a set of perturbations to choose from to maximize the chance of attacking deep neural network models. Specifically, we embrace the attacker's perspective and introduce a theoretical bound quantifying how much the universal perturbations are able to fool a given model on unseen examples. An extension to assert the transferability of universal attacks is also provided. To learn such perturbations, we devise an algorithmic solution with convergence guarantees under Lipschitz continuity assumptions. Moreover, we demonstrate how it can improve the performance of state-of-the-art gradient-based universal perturbation. As evidenced by our experiments, these novel universal perturbations result in more interpretable, diverse, and transferable attacks.

## 1 Introduction

Embedded technologies using artificial Neural Networks (NN) are increasingly present in our daily lives. Their high expressive power has shown great success in various complex tasks (Miotto et al., 2018; Grigorescu et al., 2020). However, some concerns have been raised about their safety and, more particularly, for the safety of the user (Huang et al., 2020) since the pioneering work of Szegedy et al. (2014) which has shown the existence of adversarial attacks. The most striking example is that of automated vehicles, where malicious attacks could lead the car to take unwanted action with dramatic consequences (Qayyum et al., 2020; Nassi et al., 2020).

Most of the adversarial attacks are quasi-negligible perturbations that fool the NN prediction. From a fast one-shot method (Goodfellow et al., 2015) to the first iterative procedures (Papernot et al., 2016; Moosavi-Dezfooli et al., 2016; Kurakin et al., 2017; Carlini & Wagner, 2017; Madry et al., 2018), the crafting of adversarial perturbations has lately received a lot of attention from the machine learning community. To this regard, momentum-based methods (Dong et al., 2018; Wang et al., 2021) have shown a promising boost in the transferability of the attacks learned on one NN to other NNs. In addition, various contributions have investigated algorithmic concerns leading to accelerated and scale-invariant attacks (Lin et al., 2020) as well as parameter-free attacks (Croce & Hein, 2020). In another line of research, Finlay et al. (2019) designed attacks exploiting the decision boundary of NNs, while Zhang et al. (2020b) proposed to take into account the structure of images through a principal component analysis. A key particularity of all the above attacks is that they are *specific* (or *example-based*), meaning that they are crafted to attack a *single* example. Henceforth, to attack a new example, one needs to learn the associated perturbation once again. Although they are very effective, they have the major drawback of being time-consuming.

On the other end of the spectrum, *universal* (or *example-agnostic*) attacks (Zhang et al., 2021) aim to find an attack that, once learned, can be applied to every new example. Moosavi-Dezfooli et al. (2017) first demonstrated that there exists a single perturbation, coined universal adversarial perturbation (UAP), which, when added to any new example, is very likely to fool the classifier; a variant exploiting the orientations of the perturbation vectors was proposed by Dai & Shu (2021). Later, Shafahi et al. (2020) devised a more efficient method by hinging on a projected gradient descent algorithm. In addition, inspired by the observation that UAP does not attack all classes equally, Benz et al. (2021) proposed a class-based universal perturbation. Although these perturbations are universal, it is hard to interpret why they work on a case-to-case basis. In general, current state-of-the-art universal attacks remain hardly interpretable out-of-the-box and require *a posteriori* tailored studies (Zhang et al., 2020a). These works have suggested that a reasonable assumption is that the

perturbations should live in a low-dimensional manifold. This assumption has been justified by Gu & Rigazio (2015), and later by Tabacof & Valle (2016). Some works proposed solutions to learn such a manifold (Khrulkov & Oseledets, 2018; Zhang et al., 2020b; Baluja & Fischer, 2018; Hayes & Danezis, 2018; Xiao et al., 2018). Finally, Zhang et al. (2021) suggest that simple gradient-based UAP methods may lead to better fooling performance.

**Contributions.** We propose an extension of universal perturbations that bridges the gap between specific and universal perturbations. This extension, which we call *generalized universal attacks*, starts from a theoretical observation: we derive a generalization bound on the deviation between the true and the empirical fooling risks of a universal attack. Put into words, we get a bound on how much a learned perturbation is able to fool new examples, confirming that we can use the learned perturbation to attack a model on data coming from the same task. While this bound stands for classical universal perturbation, it can be generalized to a set of universal perturbations. From this theoretical result, given a set of $L$ different *universal* perturbations, we propose introducing *generalized universal attacks* as follows. The original idea is to *specifically* craft an attack for each example by choosing, in an unsupervised manner, a perturbation among a set of $L$ different *universal* perturbations. To do so, we define an optimization problem to jointly learn the $L$ perturbations allowing each example to choose its own perturbation; $L$ can be seen as a tuning parameter controlling the amount of diversity between the perturbations. To solve it, we derive a gradient-based solution with convergence guarantees. We then propose a simple attack procedure. Lastly, our experiments confirm the effectiveness of the generalized universal perturbations over previous gradient-based UAP, in line with the conclusion of Zhang et al. (2021). Our results additionally show that they lead to more interpretable and transferable attacks.

**Outline.** We recall the general framework of adversarial perturbations in Section 2. In Section 3, after giving a generalization bound for classical universal attacks, we state our main theoretical result on which we build our *generalized universal perturbations* in Section 4: a new method to learn the perturbations and a method for attacking unseen examples. Before concluding, we conduct experiments in Section 5 on benchmark datasets. Note that we defer the proofs in the appendix.

## 2 PRELIMINARIES AND RELATED WORKS

We stand in a multiclass setting where $\mathcal{X} \subseteq \mathbb{R}^P$ is a $P$-dimensional input space and $\mathcal{Y} = \{1, \ldots, c\}$ is the set of $c \in \mathbb{N}_+$ classes. We consider an unknown data distribution $D$ on $\mathcal{X} \times \mathcal{Y}$ that models the task; the associated marginal distribution on $\mathcal{X}$ is $D_{\mathcal{X}}$. We denote $D^n$, *resp.* $D_{\mathcal{X}}^n$, the distribution of a sample constituted of $n$ data points *i.i.d.* sampled from $D$, *resp.* $D_{\mathcal{X}}$. As an attacker, we consider that we have at our disposal a *trained* model $f \colon \mathbb{R}^P \to \mathbb{R}^c$ which associates each example[1] $x \in \mathcal{X}$ to its probabilities $f(x) \in \mathbb{R}^c$ to belong to any of the $c$ classes from $\mathcal{Y}$; we denote by $\mathcal{F}$, the set of possible such models. The predicted class of $x$ by $f$ is then defined as $C_f(x) = \arg\max_{y \in \mathcal{Y}} f(x)_y$. We aim to find for each *original example* $x \sim D_{\mathcal{X}}$ a point $a \in \mathcal{X}$ close to $x$ such that $C_f(a) \neq C_f(x)$. Since $a$ is close to $x$, one can expect $f$ to predict the same class for both examples. Thus, $a$ is called *adversarial example* and said to fool the classifier $C_f$.

There exists a vast literature on the ways to build adversarial examples, measure their closeness to original examples, and quantify how much they affect the decision process of $f$. Here, we embrace the common setting of adversarial example $a = x + \varepsilon$ built by adding an *adversarial perturbation* $\varepsilon$ to an original example $x$. We consider that the two are close if the $\ell_p$-norm of the added perturbation is small enough (Laidlaw et al., 2021), *i.e.*, $\|\varepsilon\|_p \leq \delta$, for some small budget $\delta > 0$.

In addition, to measure the discrepancy of the predictions between an original sample and its perturbed version, we consider a loss function $H \colon \mathbb{R}^c \times \mathcal{Y} \to \mathbb{R}$ taking as inputs $f(a)$ and a class $k$ from $\mathcal{Y}$ that is either $C_f(x)$ or the original class $y$ of $x$. In this paper, we use the cross-entropy loss (or its $[0, 1]$-bounded counterpart (Dziugaite & Roy, 2018)).

Given the model $f$, and an unlabeled data set $S_{\mathcal{X}} = \{x_i\}_{i=1}^n \sim D_{\mathcal{X}}^n$, as an attacker, we have the possibility to craft adversarial perturbations with two current paradigms: *(i) specific attacks*, where for each $x \in S_{\mathcal{X}}$ we look for a perturbation $\varepsilon(x) \in \mathcal{B}_p(\delta) = \{e \in \mathbb{R}^P \mid \|e\|_p \leq \delta\}$, specifically tailored to attack the example $x$ (hereafter, we drop the dependency on $x$ and simply denote $\varepsilon$);

---

[1] We make the distinction between $\mathcal{X}$ and $\mathbb{R}^P$ since the input data can live inside a manifold (*e.g.*, the space of images whose pixels' intensity lies within $[0,1]$).

*(ii) universal attacks*, where we look for a perturbation $\varepsilon$ such that $a = x + \varepsilon$ is an adversarial example for all $x$ from $S_{\mathcal{X}}$. To learn such adversarial perturbations, we assume that we have a labeled sample $S = \{(x_i, y_i)\}_{i=1}^n \in (\mathcal{X} \times \mathcal{Y})^n$ consisting of $n$ data points (where the classes are either the true ones or the ones predicted by the classifier $C_f$). We recall below the two most popular specific attacks.

**DeepFool** (Moosavi-Dezfooli et al., 2016). For a given $x \in S_{\mathcal{X}}$, the DeepFool attack is the smallest $\ell_p$ perturbation managing to fool the classifier $C_f$. More formally, it solves $\text{minimize}_{\varepsilon \in \mathbb{R}^P} \|\varepsilon\|_p$ s.t. $C_f(x + \varepsilon) \neq C_f(x)$.

**PGD** (Madry et al., 2018). Given $S$, and a loss function $H$, the adversarial perturbation for a given $(x, y) \in S$ is defined as the one inside the $\ell_p$-ball which maximizes the loss between $x + \varepsilon$ and $y$, *i.e.*,

$$\underset{\varepsilon \in \mathbb{R}^P}{\text{maximize}} \; H(f(x + \varepsilon), y) \quad \text{s.t.} \quad \|\varepsilon\|_p \leq \delta. \tag{1}$$

In practice, the opposite of the objective in Equation (1) is minimized by resorting to a projected gradient method, hence the name of the attack.

We now turn to the other side of the spectrum and present some popular universal attacks.

**UAP** (Moosavi-Dezfooli et al., 2017). The first universal perturbation $\varepsilon$ was crafted by aggregating the DeepFool perturbations $\Delta \varepsilon_i$ associated to all samples $x_i \in \{x \in S_{\mathcal{X}} \mid C_f(x + \varepsilon) = C_f(x)\}$. The aggregation step is defined as $\varepsilon \leftarrow \text{Proj}_{\mathcal{B}_p(\delta)}(\varepsilon + \Delta \varepsilon_i)$, with $\text{Proj}_{\mathcal{B}_p(\delta)}$ being the projection onto $\mathcal{B}_p(\delta)$, and is repeated until some fooling rate is reached.

**Fast-UAP** (Dai & Shu, 2021). This attack relies on UAP but, instead of aggregating all the perturbations $\Delta \varepsilon_i$, it only adds the perturbation with the closest orientation to the current iterate $\varepsilon$.

**UAP-PGD** (Shafahi et al., 2020). Given $S$, the UAP-PGD attack elaborates upon PGD by framing the universal perturbation as the solution of the following optimization problem

$$\underset{\varepsilon \in \mathbb{R}^P}{\text{maximize}} \; \frac{1}{n} \sum_{(x_i, y_i) \in S} H(f(x_i + \varepsilon), y_i) \quad \text{s.t.} \quad \|\varepsilon\|_p \leq \delta. \tag{2}$$

**CW-UAP** (Benz et al., 2021). Recently, UAP-PGD has been extended to class-wise UAP, where a universal perturbation is built for each class. Let us denote $\forall k \in \mathcal{Y}$, $S_k = \{x_i \mid (x_i, k) \in S\}$ the set of training points of the $k$-th class and $n_k$ the size of $S_k$, then CW-UAP aims at solving

$$\underset{\{\varepsilon_k \in \mathbb{R}^P\}_{k \in \mathcal{Y}}}{\text{maximize}} \; \sum_{k \in \mathcal{Y}} \frac{1}{n_k} \sum_{x_i \in S_k} H(f(x_i + \varepsilon_k), k) \quad \text{s.t.} \quad \forall k \in \mathcal{Y}, \|\varepsilon_k\|_p \leq \delta. \tag{3}$$

The solution amounts to learning multiple independent UAP-PGD perturbations, one for each class.

## 3    GENERALIZATION GUARANTEES: WE CAN ATTACK UNSEEN EXAMPLES

### 3.1    FROM UNIVERSAL PERTURBATION...

In this section, we are interested in giving guarantees on the performance of the learned perturbation in the case of universal perturbations. To do so, we derive below *generalization bounds* (see, *e.g.*, Mohri et al., 2012) on the performance of the learned universal perturbation on unseen examples. Formally, given the model $f : \mathbb{R}^P \to \mathbb{R}^c$ (learned from a dataset different from $S$), and a loss function $H : \mathbb{R} \times \mathcal{Y} \to [0, 1]$, we are interested in the *quality* of the learned perturbation $\varepsilon \in \mathcal{B}_p(\delta)$ on the new examples that we measure with the *true fooling risk* defined by $R_D^f(\varepsilon) = \mathbb{E}_{(x,y) \sim D} H(f(x + \varepsilon), y)$. Since $D$ is unknown, we have no access to its value, but we can compute its empirical counterpart on the labeled dataset $S \sim D^n$ that we call *empirical fooling risk* and defined by $R_S^f(\varepsilon) = \frac{1}{n} \sum_{i=1}^n H(f(x_i + \varepsilon), y_i)$. Note that these definitions extend the fooling rate, for which $H$ is the 0-1 loss. Since our objective is to learn perturbation that fools the model, our goal is to maximize the risk. To ensure that $R_S^f(\varepsilon)$ is a good estimator of $R_D^f(\varepsilon)$, we prove the following theorem based on the Rademacher complexity (Bartlett & Mendelson, 2002).

**Proposition 1.** *For any distribution $D$ on $\mathcal{X} \times \mathcal{Y}$, for any loss function $H : \mathbb{R}^c \times \mathcal{Y} \to [0, 1]$, for any model $f : \mathbb{R}^P \to \mathbb{R}^c$, for any budget $\delta > 0$, for any $\ell_p$-norm with $p \geq 0$ and, for any $\lambda \in (0, 1]$, we have*

$$\underset{S \sim D^n}{\mathbb{P}} \left( \forall \varepsilon \in \mathcal{B}_p(\delta), \left| R_D^f(\varepsilon) - R_S^f(\varepsilon) \right| \leq 2 \Re_S [\mathcal{B}_p(\delta)] + 3 \sqrt{\frac{\ln \frac{4}{\lambda}}{2n}} \right) \geq 1 - \lambda, \tag{4}$$

*where we define the Rademacher complexity (Bartlett et al., 2002) of $\mathcal{B}_p(\delta)$ as*

$$\mathfrak{R}_S\left[\mathcal{B}_p(\delta)\right] = \underset{\boldsymbol{\sigma} \sim \Sigma^n}{\mathbb{E}} \sup_{\varepsilon \in \mathcal{B}_p(\delta)} \left[\frac{1}{n}\sum_{i=1}^n \sigma_i H(f(x_i+\varepsilon), y_i)\right], \tag{5}$$

*with $\boldsymbol{\sigma} = \{\sigma_i\}_{i=1}^n \sim \Sigma^n$, and $\Sigma$ being the Rademacher distribution, i.e., $\Sigma(-1) = \frac{1}{2}$ and $\Sigma(+1) = \frac{1}{2}$.*

Equation (4) tells that for the perturbation $\varepsilon \in \mathcal{B}_p(\delta)$, the empirical risk $R_S^f(\varepsilon)$ does not deviate too much from the true risk $R_D^f(\varepsilon)$ when the Rademacher complexity $\mathfrak{R}_S[\mathcal{B}_p(\delta)]$ is small. Note that Proposition 1 is valid for any perturbation $\varepsilon$ that lives in $\mathcal{B}_p(\delta)$ (whatever the budget $\delta$ and the $p$-norm). As an attacker, we are mostly interested in the lower bound on $R_D^f(\varepsilon)$ that gives an estimate of the "chances" to fool the model. Put into words, the more the learned perturbation manages to fool the model $f$ on $S$, the higher the empirical fooling risk, and the higher the chances to fool the model on new examples coming from the unknown distribution $D$.

The form of the bound of Proposition 1 is quite classical, but it has the originality to state a theoretical certification from the attacker's point of view to estimate—given a model $f$—the true fooling risk $R_D^f(\varepsilon)$ to quantify *how much the learned perturbations are able to fool a given model on unseen examples*. Indeed, in the literature, many recent works aim to understand the generalization abilities in an adversarial setting, but they take the defender's point of view and provide generalization bounds for the so-called *true adversarial risk* that measures *how much the learned model is able to face adversarial attacks on unseen examples*. In other words, while we study the fooling abilities of the perturbations for a given model, they study the performance (or robustness) of the model when this latter is attacked by adversarial perturbations (without knowing the attack). Among these works, we can mention Yin et al. (2019); Khim & Loh (2018); Awasthi et al. (2020) that are based on an *adversarial* Rademacher complexity of the class of the models. Other generalization bounds have been derived for the adversarial risk, such as with VC-dimension (Attias et al., 2022; Montasser et al., 2019), with covering numbers (Mustafa et al., 2022), with algorithmic stability (Xing et al., 2021), with perturbation analysis (Zeng et al., 2023), or in PAC-Bayes (Viallard et al., 2021).

### 3.2 . . . TO GENERALIZED UNIVERSAL PERTURBATIONS

From the attacker's viewpoint, learning only one perturbation $\varepsilon \in \mathcal{B}_p(\delta)$ may be inefficient: the perturbation may not fool the model for every example in the data set $S$. More formally, given only one perturbation $\varepsilon \in \mathcal{B}_p(\delta)$, the attacker may have difficulties to increase the empirical fooling risk $R_S^f(\varepsilon)$. To increase the chance of fooling the model, we propose considering $L \in \mathbb{N}_+$ perturbations $\boldsymbol{\varepsilon} = [\varepsilon_1, \ldots, \varepsilon_L] \in \mathcal{B}_p(\delta)^L$ and, for each pair $(x_i, y_i) \in S$, picking one of the $L$ perturbations in $\boldsymbol{\varepsilon}$ which maximizes the loss between $f(x_i+\varepsilon_l)$ and $y_i$ the most. In other words, the perturbations are specific enough for each example while being sufficiently universal to cover all the pairs in $S$. Hence, considering $L$ perturbations may be better suited to fool all the examples (as later shown in Section 5) since it can increase our new notion of empirical and true fooling risk defined respectively as $R_S^f(\boldsymbol{\varepsilon}) = \frac{1}{n}\sum_{i=1}^n \max_{\varepsilon_l \in \boldsymbol{\varepsilon}} H(f(x_i+\varepsilon_l), y_i)$ and $R_D^f(\boldsymbol{\varepsilon}) = \mathbb{E}_{(x,y) \sim D} \max_{\varepsilon_l \in \boldsymbol{\varepsilon}} H(f(x+\varepsilon_l), y)$. We now provide another generalization bound in the next theorem that extends Proposition 1 to the case of a set of $L$ universal perturbations.

**Theorem 2.** *Given the assumptions of Proposition 1, then for any $L \in \mathbb{N}_+$, we have*

$$\underset{S \sim D^n}{\mathbb{P}}\left(\forall \boldsymbol{\varepsilon} \in \mathcal{B}_p(\delta), \left|R_D^f(\boldsymbol{\varepsilon}) - R_S^f(\boldsymbol{\varepsilon})\right| \leq 2\,\mathfrak{R}_S\left[\mathcal{B}_p(\delta)^L\right] + 3\sqrt{\frac{\ln\frac{4}{\lambda}}{2n}}\right) \geq 1 - \lambda, \tag{6}$$

$$\text{where} \quad \mathfrak{R}_S\left[\mathcal{B}_p(\delta)^L\right] = \underset{\boldsymbol{\sigma} \sim \Sigma^n}{\mathbb{E}} \sup_{\boldsymbol{\varepsilon} \in \mathcal{B}_p(\delta)^L} \left[\frac{1}{n}\sum_{i=1}^n \sigma_i \max_{\varepsilon_l \in \boldsymbol{\varepsilon}} H(f(x_i+\varepsilon_l), y_i)\right]. \tag{7}$$

Equation (6) is a direct extension of the previous result: Proposition 1 is a special case of Theorem 2 when $L = 1$. Note that his bound is valid for any perturbation $\varepsilon$ that lives in $\mathcal{B}_p(\delta)^L$ with $L \in \mathbb{N}_+$. Importantly, Equation (6) holds for any model $f$ we want to attack that is not necessarily the one we used to learn $\varepsilon$. Hence, the bound holds in the context of adversarial transferability, where attacks

---

**Algorithm 1** $L$-UAP

---

**Require:** Parameter $\rho \in ]0, 1[$
   Initialize $\boldsymbol{\varepsilon}^{(0)} = [\varepsilon_l^{(0)}]_{l=1}^L$
   **for** $k = 0$ to $K - 1$ **do**
      Provide a rough estimate of $\gamma_k > 0$
      *Projected gradient step:* $\boldsymbol{\varepsilon}^{(k+1/2)} = \text{Proj}_{\mathcal{B}_p(\delta)^L}(\boldsymbol{\varepsilon}^{(k)} + \gamma_k \nabla R_S^f(\boldsymbol{\varepsilon}^{(k)}))$
      *Choice of relaxation parameter*
      $i_k = 0$
      **repeat**
         $\boldsymbol{\varepsilon}^{(k+1)} = (1 - \rho^{i_k})\boldsymbol{\varepsilon}^{(k)} + \rho^{i_k}\boldsymbol{\varepsilon}^{(k+1/2)}$
         $i_k = i_k + 1$
      **until** $R_S^f(\boldsymbol{\varepsilon}^{(k+1)}) \geq R_S^f(\boldsymbol{\varepsilon}^{(k)}) + \rho^{i_k-1}h^{(k)}(\boldsymbol{\varepsilon}^{(k+1/2)})$
   **end for**
   **return** $\boldsymbol{\varepsilon}^{(K)}$

---

are typically learned on simpler surrogate models in view of being used to attack other target models (see (Qin et al., 2022) and references therein). More formally, given a target model $f'$ whose weights are not available, we aim to learn a surrogate model $f \in \mathcal{F}$ in order to craft adversarial perturbations $\boldsymbol{\varepsilon}$ for each example $x \in D_{\mathcal{X}}$ that hopefully also fool the target model $f'$. In this context, we better consider the learned model $f \in \mathcal{F}$, as shown in the following proposition.

**Proposition 3.** *Given $\mathcal{F}$ the set of possible models and the assumptions of Theorem 2, then we have*

$$\mathbb{P}_{S \sim D^n}\left(\forall f \in \mathcal{F}, \boldsymbol{\varepsilon} \in \mathcal{B}_p(\delta)^L, \left|R_D^{f'}(\boldsymbol{\varepsilon}) - R_S^f(\boldsymbol{\varepsilon})\right| \leq \sup_{\boldsymbol{\varepsilon}' \in \mathcal{B}_p(\delta)^L}\left|R_S^{f'}(\boldsymbol{\varepsilon}') - R_S^f(\boldsymbol{\varepsilon})\right| + \sqrt{\frac{\ln\frac{2}{\delta}}{2n}}\right) \geq 1 - \lambda. \quad (8)$$

Equation (8) tells that the empirical risk $R_S^f(\boldsymbol{\varepsilon}')$ of our learned model $f$ is representative of the true risk $R_S^{f'}(\boldsymbol{\varepsilon}')$ of a target model $f'$ when $\sup_{\boldsymbol{\varepsilon}'}|R_S^{f'}(\boldsymbol{\varepsilon}') - R_S^f(\boldsymbol{\varepsilon})|$ is small. Intuitively, this term is small if we cannot find a set of perturbations $\boldsymbol{\varepsilon}'$ that differs too much from the perturbations $\boldsymbol{\varepsilon}$ for $f'$.

## 4 OPTIMIZATION AND SELECTION OF UNIVERSAL PERTURBATIONS

Motivated by Theorem 2, to learn a set of $L \in \mathbb{N}_+$ universal adversarial perturbations $\{\varepsilon_1, \dots, \varepsilon_L\}$, we propose to maximize the empirical fooling risk under the constraint that each $\varepsilon_l \in \mathcal{B}_p(\delta)$.

**Problem 1.** *Let $L$ be the number of universal perturbations to learn. Given $S = \{(x_i, y_i)\}_{i=1}^n$, and the model $f$, find $\boldsymbol{\varepsilon} = [\varepsilon_1, \dots, \varepsilon_L]$ solving*

$$\underset{\boldsymbol{\varepsilon} \in \mathcal{B}_p(\delta)^L}{\text{maximize}} \left\{ R_S^f(\boldsymbol{\varepsilon}) := \frac{1}{n}\sum_{i=1}^n \max_{\varepsilon_l \in \boldsymbol{\varepsilon}} H(f(x_i + \varepsilon_l), y_i) \right\}, \quad (9)$$

When $L = 1$ (*i.e.*, for a single perturbation) Equation (9) boils down to Equation (2). In addition, it bears similarities with Equation (3) when $L$ equals the number of classes and each $\varepsilon_l$ is independently learned on $S_l$. Due to the model $f$, it is worth stressing that Problem 1 is a difficult non-concave maximization problem. Finding its global solution is thus out of reach. To tackle this challenge, we embrace a projected gradient ascent algorithm augmented with an Armijo-like line-search strategy to efficiently find an approximate solution. The corresponding algorithmic procedure is sketched in Algorithm 1 while details are reported in Appendix A.1. Note that we defer in Appendix A.2 an alternative solution based on a stochastic solver fully exploiting the finite-sum nature of Problem 1. Algorithm 1 comes with convergence guarantees stated below.

**Theorem 4** (Convergence (Bonettini et al., 2017))**.** *Let $\{\boldsymbol{\varepsilon}^{(k)}\}_{k \in \mathbb{N}}$ be the sequence of Algorithm 1. Suppose that $\nabla R_S^f$ is Lipschitz continuous. Then each limit point of $\{\boldsymbol{\varepsilon}^{(k)}\}_k$ is a stationary point of Problem 1 and $\{R_S^f(\boldsymbol{\varepsilon}^{(k)})\}_k$ converges towards the objective value at the limit point. If $R_S^f$ satisfies the Kurdyka-Łojasiewicz (KŁ) property at any point, then the sequence converges to a stationary point of Problem 1.*

The presence of a Lipschitz constant is crucial to ensure convergence guarantees. Although most common loss functions $H$ for classification-based neural networks lack Lipschitz continuous gradients, variants that do exist. Hereafter, we resort to one of them, namely the bounded cross-entropy Dziugaite & Roy (2018) where the probabilities are bounded away from 0 and 1. This loss comes particularly useful as it can be rescaled to yield values in $[0, 1]$ as required by our Theorem 2. Note that studying the Lipschitz continuity of NN and obtaining sharp Lipschitz constant is difficult (*e.g.*, (Combettes & Pesquet, 2020; Gouk et al., 2021)).

**Remark 1.** *Many functions met in NNs (e.g., activation functions, loss) are semi-algebraic or tame, and thus, satisfy the KŁ property (Attouch et al., 2011; Zeng et al., 2019). Since these concepts are stable under many operations, it is reasonable to assume that many deep NNs $f$ are likely to satisfy the KŁ property and so does $R_S^f$.*

While little attention is usually devoted to these concerns for crafting adversarial attacks, we empirically show in Section 5 the superiority of the corresponding principled algorithmic solution even though these assumptions do not always hold.

According to Theorem 2, once $\varepsilon$ has been learned with Algorithm 1 from a sample $S \sim D^n$, we can attack a new example by picking one perturbation amongst the $L$ perturbations as follows.

**Problem 2** (Attacking unseen example)**.** *Given a data pair $(x, y) \sim D$, given a model $f$, the associated adversarial attack reads $a = \mathrm{Proj}_{\mathcal{X}}(x + \hat{\varepsilon})$ where $\hat{\varepsilon} = \arg\max_{\varepsilon_l \in \varepsilon} H(f(x + \varepsilon_l), y)$. If we do not have access to $y$, and assuming that $f$ is a well-performing model, then $y$ is replaced by $C_f(x)$.*

When $f$ is a NN, in order to evaluate which perturbation from $\varepsilon = [\varepsilon_1, \dots, \varepsilon_L]$ maximizes the loss function, solving Problem 2 requires performing $L$ independent forward passes through $f$. Note that, since they are independent, they can be performed in parallel to accelerate the computation. We provide below a complexity comparison between specific and (generalized) universal attacks.

**Remark 2.** *Given a model $f$ that is a neural network whose forward complexity is of $O(d)$ for a single input sample, then the complexity to compute $\nabla H(f(x), y)$ is of order $O(2d)$, since the backward pass is also of order $O(d)$. Then, it follows that* (specific) *for $K$ iterations,* $\mathrm{cost} \sim O(2Kd)$*; and* (generalized-universal) *for $L$ perturbations,* $\mathrm{cost} \sim O(Ld)$*;* (universal) $\mathrm{cost} \sim O(1)$*.*

Hence, from the standpoint of computational complexity, universal attacks are the most efficient. To a lesser extent, one-shot specific attacks (*i.e.*, $K = 1$, such as in FGSM (Goodfellow et al., 2015)) and our proposed generalized universal attack achieve comparable complexity for small $L$.

## 5 NUMERICAL EXPERIMENTS

We now conduct experiments on 3 popular benchmark classification datasets and 2 NN architectures: a differentiable multi-layer perceptron (MNIST, LeCun & Cortes, 2010) and ResNets (CIFAR-10, Krizhevsky & Hinton (2009) and ImageNet). We consider $\ell_\infty$-attacks with a maximum budget $\delta = 8/255$. For reproducibility purposes, we report implementation details such as pre-processing, data splitting, and model tuning in the supplementary material, as well as results on $\ell_2$-attacks.

### 5.1 MNIST EXPERIMENTS

We begin with the simple yet interesting MNIST dataset useful to interpret the learned perturbations. **Illustration and role of the universal perturbations.** We report in Figure 1 (left) the learned universal perturbations of 5-UAP. Interestingly, they all exhibit strong patterns. In particular, we observe that $\varepsilon_1$ and $\varepsilon_5$ are very similar up to the sign difference. Indeed, since our framework does not handle the tuning of the sign of the perturbation, it might happen that 2 perturbations are the opposite of each other. It is worth noticing that the universal perturbations learned are consistent throughout multiple splits and random initializations. We report in Figure 1 (right) the fooling matrices associated to each of the perturbations $\{\varepsilon_l\}_{l=1}^5$. The latter shows the correspondence between the predicted target $C_f(x)$ of some image $x$ (in lines) and the label of the associated adversarial attack (in columns), *i.e.*, $C_f(x + \hat{\varepsilon})$ (see Problem 2). The fooling matrices highlight that each universal perturbation plays a different role. For instance, $\varepsilon_1$ mostly allows to attack images of digits '3' and '9' to be misclassified as '5' and '4', respectively. Instead, $\varepsilon_3$ is mainly used to attack images of '5' into '3'. Coincidentally, we can distinguish the tilted digit '3' in $\varepsilon_3$. As opposed to CW-UAP, our

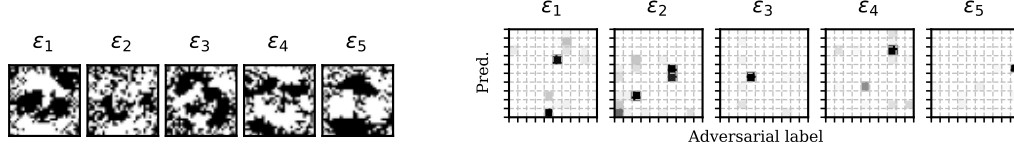

Figure 1: **5-UAP attacks on MNIST**. (Left panel) Learned adversarial perturbations. (Right panel) In the fooling matrix, labels range from 0 to 9 from top to bottom and from left to right.

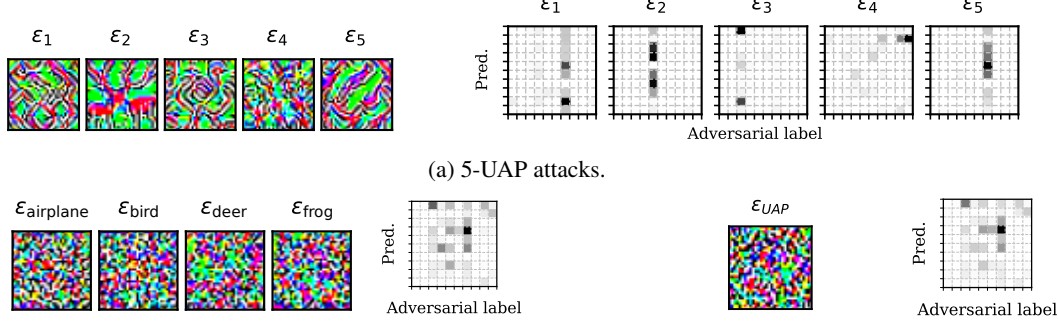

(a) 5-UAP attacks.

(b) CW-UAP attacks. Each row of the fooling matrix indicates the adversarial label found for each 10 attacks.

(c) UAP attack.

Figure 2: $\ell_\infty$-**based attacks on CIFAR-10**. In the fooling matrices, labels range {airplane, automobile, bird, cat, deer, dog, frog, horse, ship, truck} from top to bottom and left to right.

UAP attack automatically captures the similarity between multiple digits such as '3' and '9'.

**Illustration of the behavior of the generalization bound.** We report in Figure 3 an estimation of the lower and upper bounds on $R_D^f(\varepsilon)$ from Equation (6). The Rademacher complexity is approximated by resorting to the maximization Algorithm 1 where $R_S^f$ is replaced by the quantity inside the sup of Equation (7). The maximum value of the objective is then averaged over multiple draws of $\sigma \sim \Sigma^n$. As expected, we observe that the empirical fooling risk increases with $L$. While the Rademacher complexity (and so the generalization gap between $R_D^f(\varepsilon)$ and $R_S^f(\varepsilon)$) increases also with $L$ it is interesting to remark that the lower bound on $R_D^f(\varepsilon)$ tends to grow, suggesting that considering generalized universal perturbations can increase the chances to fool a model.

### 5.2 CIFAR-10 EXPERIMENTS

We turn to CIFAR-10 dataset and compare the performance of our attack with the baselines.

**Baselines.** Our $L$-UAP attack is brought into comparison against the following universal attacks[2].
• We compare with the UAP-PGD proposed by Shafahi et al. (2020) which is closely related to our 1-UAP with a single perturbation, but differs from two aspects. First, the authors have considered a capped loss with parameter $\beta$ to prevent any single sample from dominating the objective (hereafter, we use the value $\beta = 9$ that was found to be the best in (Shafahi et al., 2020)). Second, the authors resort to the stochastic normalized gradient method ADAM to learn the perturbation. Since their code is not publicly available, we tried to reproduce their version as closely as possible.
• For the sake of consistency, we implemented a Pytorch version of FAST-UAP (Dai & Shu, 2021) originally designed for TensorFlow. We use the same hyper-parameters: a desired fooling rate of $80\%$, a maximum of 10 iterations for DeepFool, an overshoot of $0.02$ to prevent vanishing updates.
• We compare against CW-UAP (Benz et al., 2021) whose code was kindly granted by the authors.
• We consider standard specific attacks such as FGSM (Goodfellow et al., 2015) and PGD (Madry et al., 2018) as well as more advanced techniques, *i.e.*, MI-FGSM (Dong et al., 2018) and AutoAttack (Croce & Hein, 2020), in order to grasp the existing gap of performance between specific and universal attacks. To this effect, we resort to the *TorchAttacks* repository (Kim, 2020).

---

[2]Pytorch codes of UAP and Fast-UAP baselines will be made publicly available along with our proposed UAP attack in order to contribute to the *TorchAttacks* repository (Kim, 2020).

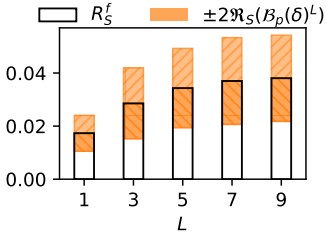

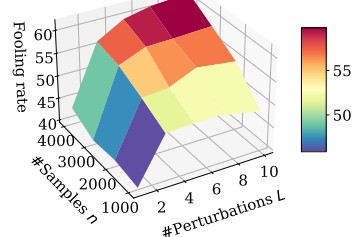

Figure 3: **Generalization bound**. The top of the orange bar, *resp.* the bottom, is the approximation of the upper, *resp.* lower, bound of $R_D^f$.

Figure 4: **Impact of parameters.** Depending on the number of CIFAR-10 samples, we report the fooling rate of $\ell_\infty$-based $L$-UAP attacks.

**Illustration & insights about UAP attacks.** Similarly to the MNIST experiment, we report in Figure 2a the learned 5-UAP universal perturbations (left) and their fooling matrices (right). We observe that each UAP universal perturbation plays a different role and illustrates the diversity in perturbations. Indeed, for instance, $\varepsilon_2$ is mostly used to attack images of animals (*bird*, *cat*, *dog*, *frog*, *horse*) so that they become misclassified as *deer*: in $\varepsilon_2$ one can distinguish two deer facing each other. Another example is $\varepsilon_3$ which is mostly used to misclassify images of *airplane* and *ship* as *bird*; in $\varepsilon_3$ one can distinguish a bird. We also report in Figures 2b and 2c the fooling matrices and some universal perturbations of the CW-UAP attack and UAP attack, respectively. Contrary to UAP, we merged all 10 fooling matrices of CW-UAP (one for each class) into a single one since they are all disjointed. Thus, each row of Figure 2b (left) corresponds to the adversarial label obtained for each of 10 independent class-wise attacks. Unsurprisingly, many of the same couples (predicted label, adversarial label) appear in the fooling matrices of UAP, CW-UAP and UAP. This makes sense since, ultimately, each couple (predicted label, adversarial label) depends on the similarity between image classes and how the classifier proceeds to distinguish between the classes. Despite this resemblance, the key point is that all three methods operate differently. Especially, by its construction $L$-UAP is able to find an overlapping decomposition of the (predicted label, adversarial label) couple. As such, it automatically unveils the similarity between examples belonging to two different classes.

**Impact of the numbers of training samples.** Herein, we take a deeper look at the impact of two parameters on the performance of UAP attacks. More precisely, we study the influence of the number of training samples $n$ and the number of perturbations $L$ (see Problem 1) on the test fooling rate. To this end, we let $n$ and $L$ vary in $\{1K, 2K, 3K, 4K\}$ and $\{1, 3, 5, 7, 10\}$, respectively. All learned $L$-UAP attacks are evaluated on a distinct test set. Results, averaged over multiple splits, are reported in Figure 4. Overall, we observe that increasing $L$ improves the performance, thus confirming that having more perturbations is beneficial to attack the network $f$. This observation has to be contrasted with the fact that the amount of data $n$ required to achieve good performance goes in pair with the complexity of the learning Problem 1, hence with $L$. As such, for $n=1K$ or $2K$, the performance does not significantly improve (or worse, decrease) with larger $L$. In what follows, we restrict to a setting made of few samples (*i.e.*, $n=2K$).

**Comparison with baselines.** We report in Table 1 the performances of $\ell_\infty$-attacks, in terms of fooling rate. First, 1-UAP outperforms all universal attacks (UAP-PGD, FAST-UAP, CW-UAP) and, most importantly, it surpasses UAP-PGD, which is closely related. We believe that this is due to our algorithm, which benefits from better optimization guarantees (see also Section E.2). In addition, as $L$ grows, we observe an increase in the performance of UAP attacks, thus justifying the advantages of having more degrees of freedom. Interestingly, the UAP attacks manage to improve upon the one-shot specific FGSM attack. However, the performances are still very far behind the more advanced specific attacks. Nonetheless, such differences in performance have to be contrasted with their associated computational complexity (see Remark 2). Overall, $L$-UAP yields a competitive trade-off between universality and specificity by tuning the number $L$ of universal perturbations.

**Transferability of attacks.** We further evaluate how the learned attacks on the ResNet18 model manage to fool more complex architectures such as the pre-trained ResNet50 and MobileNetv2 (Sandler et al., 2018) models. We additionally consider two robust models from the *RobustBench* repository (Croce et al., 2020), namely r-ResNet18 (Sehwag et al., 2022) and r-ResNet50 (Chen et al.,

Table 1: **Performance and transferability of $\ell_\infty$-attacks on a ResNet18 trained on CIFAR-10.** Results are divided into universal (top), proposed (middle) and specific (bottom) attacks. Bold fonts highlight the best fooling rate in each attack category for each target model (along the columns).

| Attack | Source | Transfer | | | |
|---|---|---|---|---|---|
| | ResNet18 | ResNet50 | MobileNetv2 | r-ResNet18 | r-ResNet50 |
| UAP-PGD (Shafahi et al., 2020) | $12.53 \pm 0.60$ | $21.51 \pm 0.18$ | $39.37 \pm 0.19$ | $2.01 \pm 0.01$ | $2.54 \pm 0.05$ |
| FAST-UAP (Dai & Shu, 2021) | $11.16 \pm 1.03$ | $19.65 \pm 1.22$ | $36.51 \pm 0.30$ | $1.94 \pm 0.01$ | $2.33 \pm 0.05$ |
| CW-UAP (Benz et al., 2021) | $\mathbf{13.85 \pm 0.18}$ | $\mathbf{21.95 \pm 0.28}$ | $\mathbf{39.62 \pm 0.33}$ | $\mathbf{2.26 \pm 0.07}$ | $\mathbf{2.26 \pm 0.06}$ |
| 1-UAP | $36.83 \pm 0.93$ | $27.45 \pm 0.81$ | $44.11 \pm 0.35$ | $2.27 \pm 0.02$ | $2.27 \pm 0.08$ |
| 3-UAP | $54.03 \pm 0.54$ | $28.49 \pm 0.56$ | $45.42 \pm 0.32$ | $2.55 \pm 0.03$ | $2.95 \pm 0.08$ |
| 5-UAP | $\mathbf{55.56 \pm 0.57}$ | $\mathbf{28.87 \pm 0.07}$ | $\mathbf{46.09 \pm 0.10}$ | $\mathbf{2.56 \pm 0.05}$ | $\mathbf{3.10 \pm 0.05}$ |
| FGSM (Goodfellow et al., 2015) | $53.82 \pm 0.00$ | $28.55 \pm 0.00$ | $38.10 \pm 0.00$ | $\mathbf{3.02 \pm 0.00}$ | $3.14 \pm 0.00$ |
| MI-FGSM (Dong et al., 2018) | $80.76 \pm 0.00$ | $29.95 \pm 0.01$ | $35.46 \pm 0.01$ | $2.60 \pm 0.00$ | $\mathbf{3.41 \pm 0.00}$ |
| PGD (Madry et al., 2018) | $\mathbf{93.61 \pm 0.06}$ | $30.47 \pm 0.12$ | $\mathbf{38.17 \pm 0.37}$ | $1.94 \pm 0.05$ | $2.53 \pm 0.08$ |
| AutoAttack (Croce & Hein, 2020) | $93.07 \pm 0.00$ | $\mathbf{31.79 \pm 0.16}$ | $38.08 \pm 0.27$ | $1.91 \pm 0.02$ | $2.41 \pm 0.02$ |

Table 2: **Performance of $\ell_\infty$-attacks on a ResNet18 trained on ImageNet.** Bold fonts highlight the best fooling rate in universal (left), proposed (middle) and specific (right) attacks.

| Attack | ResNet18 | Attack | ResNet18 | Attack | ResNet18 |
|---|---|---|---|---|---|
| UAP-PGD (Shafahi et al., 2020) | $\mathbf{27.36 \pm 0.00}$ | 1-UAP | $83.17 \pm 2.62$ | FGSM (Goodfellow et al., 2015) | $84.53 \pm 0.05$ |
| FAST-UAP (Dai & Shu, 2021) | $23.46 \pm 0.25$ | 5-UAP | $\mathbf{88.98 \pm 1.06}$ | MI-FGSM (Dong et al., 2018) | $90.04 \pm 0.02$ |
| | | 10-UAP | $87.24 \pm 1.16$ | PGD (Madry et al., 2018) | $\mathbf{94.99 \pm 0.06}$ |
| | | | | AutoAttack (Croce & Hein, 2020) | $88.23 \pm 0.05$ |

2020), which are trained with some defense mechanisms against $\ell_\infty$-attacks of budget $\delta = 8/255$. Results are reported in Table 1. Overall, $L$-UAP systematically yields better transferability than all universal attacks, as shown by the higher fooling rates. In addition, it manages to outperform specific attacks when the target model architecture is significantly different than the base model on which the attacks have been learned (*i.e.*, Mobilenetv2 vs. ResNet18). Note that this is precisely the setting where most universal attacks also show greater transferability than specific attacks. Interestingly, $L$-UAP shows competitive results on robust models. It is important to remark that our experimental results are in line with the theory proposed in Proposition 3, which shows that choosing two different architectures may not have an influence on the fooling rate as long as the perturbations similarly fool the learned and the target models.

### 5.3 IMAGENET EXPERIMENTS

We tackle a large-scale scenario made of 1K classes. Such a setting is problematic for CW-UAP as computing or storing 1K perturbations exceeds most memory storage spaces: it is not studied here.
**Results.** We report the performances in Table 2. Again, we observe a drastic gap in performance between UAP-PGD and 1-UAP, confirming the superiority of the numerical solution of Algorithm 1 for $L = 1$ perturbation over the standard UAP-PGD solver (Shafahi et al., 2020). Overall, UAP achieves performance of the order of magnitude as specific attacks (*e.g.*, MI-FGSM). It suggests that, for large-scale settings with numerous classes, solely a few universal perturbations are enough to attack most of the images.

## 6 CONCLUSION

We have established a theoretical foundation for attackers by deriving a generalization bound that quantifies the effectiveness of universal attacks on new examples and to other neural network models. This bound not only applies to classical universal perturbations but also extends to our novel generalized universal perturbations. The latter stands halfway between specific and universal attacks, as evidenced by our numerical experiments. Beyond the gain in performance, generalized universal attacks pull out of existing attacks by capturing meaningful patterns describing the most common flaws to fool the classifier. We believe that the latter might help to shed some light on how the classifier operates.

**Ethic statement.** Although this work aims at attacking DNNs, the weaknesses found could be useful for more positive goals, *e.g.*, improving the robustness of DNNs. This contributes to enhance development towards safer and more reliable DNNs.

**Reproducibility statement.** In order to contribute to more reproducible research, a PyTorch implementation of our proposed novel attack is given in the supplementary material and will be made publicly available as part of the TorchAttacks repository (Kim, 2020). In addition, the models are shared in our supplementary material as well. To this regard, we have provided the parameters of the MNIST classification model (Cireşan et al., 2010), have resorted to public CIFAR10 models from (Phan, 2021) and RobustBench (Croce et al., 2020), and have used standard models from the Torchvision repository for ImageNet experiments. Pre-processing steps are discussed in Section D. We would like to stress that no sophisticated seed selection has been conducted for performing data splitting or finding the best algorithm initialization. As such, all experimental settings described in Section D can be reproduced for seeds ranging from 0 to 4. Concerning theoretical results, complete proofs are deferred in the appendix in Section B and Section C.

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

# A    ALGORITHMIC SOLUTIONS

In this section, we detail the algorithmic procedures used to learn the generalized universal perturbations.

**Notations.** For $x \in \mathbb{R}^P$ and $\mathcal{I} \subseteq \{1, \ldots, P\}$, $x_{\mathcal{I}}$ stands for the restriction of $x$ to the indices in $\mathcal{I}$.

## A.1    DETERMINISTIC SOLVER

To maximize $R_S^f$ in Problem 1, we embrace a projected gradient ascent algorithm augmented with an Armijo-like line-search strategy (for ensuring some sufficient increase at each iteration). Its principle is inspired from the minorize-maximization algorithm where, at each step, a lower bound of the empirical fooling risk $R_S^f$ is maximized. Let $\varepsilon = [\varepsilon_1, \ldots, \varepsilon_L]$ and some sequence of step-sizes $\{\gamma_k\}_{k \in \mathbb{N}_+}$. Then, at each iteration $k \in \mathbb{N}_+$ the algorithm looks for $\varepsilon^{(k+1/2)} \in \mathcal{B}_p(\delta)^L$ which maximizes the linearized surrogate $h^{(k)}(\varepsilon) = R_S^f(\varepsilon^{(k)}) + \left\langle \nabla R_S^f(\varepsilon^{(k)}), \varepsilon - \varepsilon^{(k)} \right\rangle - (1/2\gamma_k)\|\varepsilon - \varepsilon^{(k)}\|^2$ with $\langle \cdot, \cdot \rangle$ the Frobenius inner product and $\|\cdot\|$ the induced norm. Such choice is motivated by the fact that, for concave and $\mu$-smooth functions $R_S^f$, and for all $\gamma_k \leq \frac{1}{\mu}$, we have $R_S^f(\varepsilon) \geq h^{(k)}(\varepsilon)$. Henceforth, we have

$$\varepsilon^{(k+1/2)} = \underset{\varepsilon \in \mathcal{B}_p(\delta)^L}{\arg\max}\, h^{(k)}(\varepsilon) = \text{Proj}_{\mathcal{B}_p(\delta)^L}\left(\varepsilon^{(k)} + \gamma_k \nabla R_S^f(\varepsilon^{(k)})\right), \tag{10}$$

which recasts into one projected gradient ascent step. Note that the differentiability of $R_S^f$ depends on the choice $H$ and on the NN $f$ to attack. For instance, for ReLu-based NN, it is likely that $\nabla R_S^f(\varepsilon^{(k)})$ is not well-defined. In that case and whenever $R_S^f$ is not differentiable, we resort to a sub-gradient instead. We additionally consider a relaxation step of the form $\varepsilon^{(k+1)} = (1 - \alpha_k)\varepsilon^{(k)} + \alpha_k\varepsilon^{(k+1/2)}$, where the relaxation parameter $\alpha_k \in (0, 1]$ is appropriately chosen by an Armijo-like line-search strategy to ensure (Bonettini et al., 2017) some sufficient increase in $R_S^f$. This algorithmic procedure is sketched in Algorithm 1 and referred as $L$-UAP. In practice, we suggest initializing the $L$ universal perturbations in a non-informative manner by randomly sampling each $\varepsilon_l^{(0)} \sim [-\delta, \delta]^P$ and additionally projecting onto the ball $\mathcal{B}_p(\delta)$.

**About the max operator.**    Note that $R_S^f$ is first and foremost not differentiable because of the max term of the objective recalled below.

$$\frac{1}{n}\sum_{i=1}^n \max_{\varepsilon_l \in \varepsilon} H(f(x_i + \varepsilon_l), y_i). \tag{11}$$

To avoid this concern, one could replace max by a smooth approximation. However, empirically, iterates almost never lie at such singularities. To evidence such finding, we have conducted an additional experiment where the absolute difference between the two largest values of $\{H(f(x_i + \varepsilon_l), y_i)\}_{\varepsilon_l \in \varepsilon}$ is computed. In particular, we have inspected the smallest absolute difference over the training set, as a function of the iterates $\varepsilon^{(k)}$ of Algorithm 1. We report in Figure 5 one representative run to learn 5-UAP on MNIST (see Section D.1 for details about the experimental setting). At every iteration, the smallest "distance to singularity" is always strictly positive, hence the iterates never lie at the discontinuities of the max term.

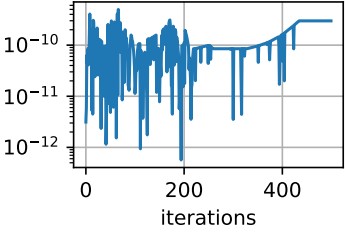

Figure 5: **Distance to singularity**.

## A.2 STOCHASTIC SOLVER

We propose an additional solver fully exploiting the finite-sum nature of the loss in Problem 1. To this regard, we begin by rewriting it by means of the sample-wise losses $r_i$, *i.e.*,

$$R_S^f(\varepsilon) = \frac{1}{n}\sum_{i=1}^{n} r_i(\varepsilon), \quad \text{with } r_i(\varepsilon) = \max_{l\in\{1,\dots,L\}} H(f(x_i + \varepsilon_l), y_i).$$

Hereafter, we resort to a stochastic solver based on the well-known variance reduction techniques (see (J. Reddi et al., 2016; Wang et al., 2019; Pham et al., 2020)). Since the main computational load comes from the backpropagation through the neural network, we favor the proxSAGA algorithm (J. Reddi et al., 2016) which does not require an additional loop over multiple epochs. The corresponding algorithmic solution is summarized in Algorithm 2.

---

**Algorithm 2** UAP-ProxSAGA

---

Initialize $\varepsilon^{(0)} = [\varepsilon_l^{(0)}]_{l=1}^{L}$
Set $g_i = \nabla r_i(\varepsilon^{(0)})$ for every $i \in \{1, \dots, n\}$
Set $\bar{g}^{(0)} = (1/n)\sum_{i=1}^{n} g_i$
**for** $k = 0$ to $K - 1$ **do**
    *Instant gradient computation*
    Uniformly pick a batch $\mathcal{I}_k \subset \{1, \dots, n\}$ of size $b$
    $g_{\mathcal{I}_k} = \sum_{i\in\mathcal{I}_k} \nabla r_i(\varepsilon^{(k)})$
    *Projected gradient step*
    $\alpha^{(k)} = \frac{1}{b}(g_{\mathcal{I}_k} - \tilde{g}_{\mathcal{I}_k}) + \bar{g}^{(k)}$
    $\varepsilon^{(k+1)} = \mathrm{Proj}_{\mathcal{B}_p(\delta)}(\varepsilon^{(k)} + \gamma_k \alpha^{(k)})$
    *Updates*
    $\bar{g}^{(k+1)} = \frac{1}{n}(g_{\mathcal{I}_k} - \tilde{g}_{\mathcal{I}_k}) + \bar{g}^{(k)}$
    $\tilde{g}_{\mathcal{I}_k} = g_{\mathcal{I}_k}$
**end for**
**return** Generalized universal adversarial perturbations $\varepsilon^{(K)}$

---

Such solver should become particularly useful to deal with large datasets by treating one sample at a time. We recall below the convergence guarantees under the assumption of Lipschitz continuity.

**Theorem 5** (J. Reddi et al. (2016)). *Suppose that $\nabla R_S^f$ is Lipschitz continuous with Lipschitz constant $\beta$. Let $\{\varepsilon^{(k)}\}_{k\in\mathbb{N}}$ be the sequence of Algorithm 2 with fixed step-size $\gamma_k = \gamma \leq 1/(5\beta n)$ and batch-size $b = 1$. Then, for $k$ uniformly sampled from $\{1, \dots, K\}$, the following holds:*

$$\mathbb{E}\left[\|G_\gamma(\varepsilon^{(k)})\|^2\right] \leq \frac{50\beta n^2}{5n-2} \frac{R_S^f(\varepsilon^\star) - R_S^f(\varepsilon^{(0)})}{K}, \tag{12}$$

*where $\varepsilon^\star$ is a maximizer of $R_S^f$ and $G_\gamma\colon \varepsilon \mapsto \gamma^{-1}(\varepsilon - \mathcal{P}_{\mathcal{B}_p(\delta)}(\varepsilon + \gamma\nabla R_S^f(\varepsilon)))$ is the gradient mapping.*

Note that Theorem 5 relies on the Lipschitz constant $\beta$ whose calculation is out of reach. Instead, in practice we suggest either to choose $\beta$ large enough or to compute rough estimate at each iteration.

# B PROOF OF PROPOSITION 1 AND THEOREM 2

The proof of Proposition 1 and Theorem 2 relies on Theorem 3.3 of Mohri et al. (2012).

**Theorem 3.3 of Mohri et al. (2012).** For any distribution $D$ on $\mathcal{X} \times \mathcal{Y}$, for any set $\mathcal{G}$ of functions $g : \mathcal{X} \times \mathcal{Y} \to [0, 1]$, for any $\lambda \in (0, 1]$, we have

$$\mathbb{P}_{S \sim D^n} \left( \forall g \in \mathcal{G}, \; \mathbb{E}_{(x,y) \sim D} g(x, y) - \frac{1}{n} \sum_{i=1}^{n} g(x_i, y_i) \right.$$
$$\left. \leq 2 \mathbb{E}_{\boldsymbol{\sigma} \sim \Sigma^n} \left[ \sup_{g' \in \mathcal{G}} \frac{1}{n} \sum_{i=1}^{n} \sigma_i g'(x_i, y_i) \right] + 3 \sqrt{\frac{\ln \frac{2}{\lambda}}{2n}} \right) \geq 1 - \lambda. \qquad (13)$$

Before proving Proposition 1 and Theorem 2, we recall how we can obtain a two-sided generalization bound from Mohri et al. (2012)'s Theorem 3.3.

Mohri et al. (2012)'s Theorem 3.3 brings a one-sided generalization bound, *i.e.*, an upper bound on $\mathbb{E}_{(x,y) \sim D} g(x, y) - \frac{1}{n} \sum_{i=1}^{n} g(x_i, y_i)$. That being said, Proposition 1 and Theorem 2 provide a two-sided bound, *i.e.*, an upper bound on the term $\left| \mathbb{E}_{(x,y) \sim D} g(x, y) - \frac{1}{n} \sum_{i=1}^{n} g(x_i, y_i) \right|$. One common solution is to use the union bound (*e.g.*, (Mehta, 2021)). For the sake of completeness, we state the two-sided bound associated with Theorem 3.3. of Mohri et al. (2012) in the following lemma.

**Lemma 1** (Two-sided generalization bounds)**.** *For any distribution $D$ on $\mathcal{X} \times \mathcal{Y}$, for any set $\mathcal{G}$ of functions $g : \mathcal{X} \times \mathcal{Y} \to [0, 1]$, for any $\lambda \in (0, 1]$, we have*

$$\mathbb{P}_{S \sim D^n} \left( \forall g \in \mathcal{G}, \; \left| \mathbb{E}_{(x,y) \sim D} g(x, y) - \frac{1}{n} \sum_{i=1}^{n} g(x_i, y_i) \right| \right.$$
$$\left. \leq 2 \mathbb{E}_{\boldsymbol{\sigma} \sim \Sigma^n} \left[ \sup_{g' \in \mathcal{G}} \frac{1}{n} \sum_{i=1}^{n} \sigma_i g'(x_i, y_i) \right] + 3 \sqrt{\frac{\ln \frac{4}{\lambda}}{2n}} \right) \geq 1 - \lambda. \qquad (14)$$

*Proof.* We can go through the exact same proof of Mohri et al. (2012)'s Theorem 3.3 but with $\frac{1}{n} \sum_{i=1}^{n} g(x_i, y_i) - \mathbb{E}_{(x,y) \sim D} g(x, y)$ instead of $\mathbb{E}_{(x,y) \sim D} g(x, y) - \frac{1}{n} \sum_{i=1}^{n} g(x_i, y_i)$. Hence, we obtain with probability at least $1 - \frac{\lambda}{2}$ over $S \sim D^n$

$$\forall g \in \mathcal{G}, \quad \frac{1}{n} \sum_{i=1}^{n} g(x_i, y_i) - \mathbb{E}_{(x,y) \sim D} g(x, y)$$
$$\leq 2 \mathbb{E}_{\boldsymbol{\sigma} \sim \Sigma^n} \left[ \sup_{g' \in \mathcal{G}} \frac{1}{n} \sum_{i=1}^{n} \sigma_i g'(x_i, y_i) \right] + 3 \sqrt{\frac{\ln \frac{4}{\lambda}}{2n}}. \qquad (15)$$

Hence, by combining Equations (13) and (15) from a union bound (and with $\lambda/2$ instead of $\lambda$), we obtain Equation (14). □

We are now ready to prove Proposition 1 and Theorem 2. Note that the proofs are based on Lemma 1.

**Proposition 1.** *For any distribution $D$ on $\mathcal{X} \times \mathcal{Y}$, for any loss function $H : \mathbb{R}^c \times \mathcal{Y} \to [0, 1]$, for any model $f : \mathbb{R}^P \to \mathbb{R}^c$, for any budget $\delta > 0$, for any $\ell_p$-norm with $p \geq 0$ and, for any $\lambda \in (0, 1]$, we have*

$$\mathbb{P}_{S \sim D^n} \left( \forall \varepsilon \in \mathcal{B}_p(\delta), \; \left| R_D^f(\varepsilon) - R_S^f(\varepsilon) \right| \leq 2 \mathfrak{R}_S \left[ \mathcal{B}_p(\delta) \right] + 3 \sqrt{\frac{\ln \frac{4}{\lambda}}{2n}} \right) \geq 1 - \lambda, \qquad (4)$$

*where we define the Rademacher complexity (Bartlett et al., 2002) of $\mathcal{B}_p(\delta)$ as*

$$\mathfrak{R}_S \left[ \mathcal{B}_p(\delta) \right] = \mathbb{E}_{\boldsymbol{\sigma} \sim \Sigma^n} \sup_{\varepsilon \in \mathcal{B}_p(\delta)} \left[ \frac{1}{n} \sum_{i=1}^{n} \sigma_i H(f(x_i + \varepsilon), y_i) \right], \qquad (5)$$

*with $\boldsymbol{\sigma} = \{\sigma_i\}_{i=1}^{n} \sim \Sigma^n$, and $\Sigma$ being the Rademacher distribution, i.e., $\Sigma(-1) = \frac{1}{2}$ and $\Sigma(+1) = \frac{1}{2}$.*

*Proof.* We will now provide an upper bound of the gap $\left| R_D^f(\varepsilon) - R_S^f(\varepsilon) \right|$. To do so, we define the set of function $\mathcal{G}$ by

$$\mathcal{G} := \left\{ g : (x,y) \mapsto H(f(x+\varepsilon), y) \mid \varepsilon \in \mathcal{B}_p(\delta) \right\}.$$

By applying Lemma 1 on the set $\mathcal{G}$, with probability at least $1 - \lambda$ over $S \sim D^n$ we have for all $\varepsilon \in \mathcal{B}_p(\delta)$

$$\left| R_D^f(\varepsilon) - R_S^f(\varepsilon) \right| \leq 2 \underset{\boldsymbol{\sigma} \sim \Sigma^n}{\mathbb{E}} \sup_{\varepsilon \in \mathcal{B}_p(\delta)} \left[ \frac{1}{n} \sum_{i=1}^n \sigma_i H(f(x_i+\varepsilon), y_i) \right] + 3\sqrt{\frac{\ln \frac{4}{\lambda}}{2n}},$$

which is the desired result. $\qquad\square$

We now prove Theorem 2 that has a similar proof than the one of Proposition 1.

**Theorem 2.** *Given the assumptions of Proposition 1, then for any $L \in \mathbb{N}_+$, we have*

$$\underset{S \sim D^n}{\mathbb{P}} \left( \forall \boldsymbol{\varepsilon} \in \mathcal{B}_p(\delta), \ \left| R_D^f(\boldsymbol{\varepsilon}) - R_S^f(\boldsymbol{\varepsilon}) \right| \leq 2\mathfrak{R}_S \left[ \mathcal{B}_p(\delta)^L \right] + 3\sqrt{\frac{\ln \frac{4}{\lambda}}{2n}} \right) \geq 1 - \lambda, \tag{6}$$

$$\text{where} \quad \mathfrak{R}_S \left[ \mathcal{B}_p(\delta)^L \right] = \underset{\boldsymbol{\sigma} \sim \Sigma^n}{\mathbb{E}} \sup_{\boldsymbol{\varepsilon} \in \mathcal{B}_p(\delta)^L} \left[ \frac{1}{n} \sum_{i=1}^n \sigma_i \max_{\varepsilon_l \in \boldsymbol{\varepsilon}} H(f(x_i+\varepsilon_l), y_i) \right]. \tag{7}$$

*Proof.* We will now provide an upper bound of the gap $\left| R_D^f(\boldsymbol{\varepsilon}) - R_S^f(\boldsymbol{\varepsilon}) \right|$. To do so, we define the set of function $\mathcal{G}$ by

$$\mathcal{G} := \left\{ g : (x,y) \mapsto \max_{\varepsilon_l \in \boldsymbol{\varepsilon}} H(f(x+\varepsilon_l), y) \mid \boldsymbol{\varepsilon} \in \mathcal{B}_p(\delta)^L \right\}.$$

By applying Lemma 1 on the set $\mathcal{G}$, with probability at least $1 - \lambda$ over $S \sim D^n$ we have for all $\boldsymbol{\varepsilon} \in \mathcal{B}_p(\delta)^L$

$$\left| R_D^f(\boldsymbol{\varepsilon}) - R_S^f(\boldsymbol{\varepsilon}) \right| \leq 2 \underset{\boldsymbol{\sigma} \sim \Sigma^n}{\mathbb{E}} \sup_{\boldsymbol{\varepsilon} \in \mathcal{B}_p(\delta)^L} \left[ \frac{1}{n} \sum_{i=1}^n \sigma_i \max_{\varepsilon_l \in \boldsymbol{\varepsilon}} H(f(x_i+\varepsilon_l), y_i) \right] + 3\sqrt{\frac{\ln \frac{4}{\lambda}}{2n}},$$

which is the desired result. $\qquad\square$

## C  PROOF OF PROPOSITION 3

In order to prove Proposition 3, we first prove the following lemma.

**Lemma 2.** *For any distribution $D$ on $\mathcal{X} \times \mathcal{Y}$, for any set $\mathcal{G}$ of functions $g : \mathcal{X} \times \mathcal{Y} \to [0,1]$, for any set $\mathcal{G}'$ of functions $g' : \mathcal{X} \times \mathcal{Y} \to [0,1]$, for any $\lambda \in (0,1]$, with probability at least $1 - \lambda$ over $S \sim D^n$ we have*

$$\underset{S \sim D^n}{\mathbb{P}} \left( \forall S' \in \mathcal{Z}^m, g \in \mathcal{G}, g' \in \mathcal{G}', \ \left| \underset{(x,y) \sim D}{\mathbb{E}} g(x,y) - \frac{1}{m} \sum_{i=1}^m g'(x_i', y_i') \right| \right.$$

$$\left. \leq \sup_{g \in \mathcal{G}} \left| \frac{1}{n} \sum_{i=1}^n g(x_i, y_i) - \frac{1}{m} \sum_{i=1}^m g'(x_i', y_i') \right| + \sqrt{\frac{\ln \frac{2}{\delta}}{2n}} \right) \geq 1 - \lambda$$

*Proof.* First of all, we have

$$\sup_{g \in \mathcal{G}} \left( \underset{(x,y) \sim D}{\mathbb{E}} g(x,y) \right) = \sup_{g \in \mathcal{G}} \left( \underset{S \sim D^n}{\mathbb{E}} \frac{1}{n} \sum_{i=1}^n g(x_i, y_i) \right) \leq \underset{S \sim D^n}{\mathbb{E}} \sup_{g \in \mathcal{G}} \left( \frac{1}{n} \sum_{i=1}^n g(x_i, y_i) \right). \tag{16}$$

Then, from McDiarmid's inequality, we have with probability at least $1 - \lambda/2$ over $S \sim D^n$

$$\mathbb{E}_{S \sim D^n} \sup_{g \in \mathcal{G}} \left( \frac{1}{n} \sum_{i=1}^{n} g(x_i, y_i) \right) \leq \sup_{g \in \mathcal{G}} \left( \frac{1}{n} \sum_{i=1}^{n} g(x_i, y_i) \right) + \sqrt{\frac{\ln \frac{2}{\delta}}{2n}}. \tag{17}$$

Hence, by combining Equations (16) and (17), we have

$$\sup_{g \in \mathcal{G}} \left( \mathbb{E}_{(x,y) \sim D} g(x, y) \right) \leq \sup_{g \in \mathcal{G}} \left( \frac{1}{n} \sum_{i=1}^{n} g(x_i, y_i) \right) + \sqrt{\frac{\ln \frac{2}{\delta}}{2n}}. \tag{18}$$

We add $\frac{1}{m} \sum_{i=1}^{m} -g'(x_i', y_i')$ to both sides of the inequality to obtain for all $S' \in \mathcal{Z}^m$ and $g' \in \mathcal{G}'$

$$\sup_{g \in \mathcal{G}} \left( \mathbb{E}_{(x,y) \sim D} g(x, y) - \frac{1}{m} \sum_{i=1}^{m} g'(x_i', y_i') \right) \leq \sup_{g \in \mathcal{G}} \left( \frac{1}{n} \sum_{i=1}^{n} g(x_i, y_i) - \frac{1}{m} \sum_{i=1}^{m} g'(x_i', y_i') \right) + \sqrt{\frac{\ln \frac{2}{\delta}}{2n}}$$

$$\leq \sup_{g \in \mathcal{G}} \left| \frac{1}{n} \sum_{i=1}^{n} g(x_i, y_i) - \frac{1}{m} \sum_{i=1}^{m} g'(x_i', y_i') \right| + \sqrt{\frac{\ln \frac{2}{\delta}}{2n}}. \tag{19}$$

We can follow the exact same steps as before, but with $\sup_{g \in \mathcal{G}} (- \mathbb{E}_{(x,y) \sim D} g(x,y))$ and $\frac{1}{m} \sum_{i=1}^{m} g'(x_i', y_i')$ instead of $\sup_{g \in \mathcal{G}} (\mathbb{E}_{(x,y) \sim D} g(x,y))$ and $\frac{1}{m} \sum_{i=1}^{m} -g'(x_i', y_i')$, to obtain with probability at least $1 - \lambda/2$ over $S \sim D^n$

$$\sup_{g \in \mathcal{G}} \left( \frac{1}{m} \sum_{i=1}^{m} g'(x_i', y_i') - \mathbb{E}_{(x,y) \sim D} g(x, y) \right) \leq \sup_{g \in \mathcal{G}} \left( \frac{1}{m} \sum_{i=1}^{m} g'(x_i', y_i') - \frac{1}{n} \sum_{i=1}^{n} g(x_i, y_i) \right) + \sqrt{\frac{\ln \frac{2}{\delta}}{2n}}$$

$$\leq \sup_{g \in \mathcal{G}} \left| \frac{1}{m} \sum_{i=1}^{m} g'(x_i', y_i') - \frac{1}{n} \sum_{i=1}^{n} g(x_i, y_i) \right| + \sqrt{\frac{\ln \frac{2}{\delta}}{2n}}. \tag{20}$$

Finally, by combining Equations (19) and (20), we have the desired result. $\qquad \square$

We are now ready to prove Proposition 3.

**Proposition 3.** *Given $\mathcal{F}$ the set of possible models and the assumptions of Theorem 2, then we have*

$$\mathbb{P}_{S \sim D^n} \left( \forall f \in \mathcal{F}, \varepsilon \in \mathcal{B}_p(\delta)^L, \left| R_D^{f'}(\varepsilon) - R_S^f(\varepsilon) \right| \leq \sup_{\varepsilon' \in \mathcal{B}_p(\delta)^L} \left| R_S^{f'}(\varepsilon') - R_S^f(\varepsilon) \right| + \sqrt{\frac{\ln \frac{2}{\delta}}{2n}} \right) \geq 1 - \lambda. \tag{8}$$

*Proof.* We will now provide an upper bound of the gap $\left| R_D^{f'}(\varepsilon) - R_S^f(\varepsilon) \right|$. To do so, we apply Lemma 2 with the fixed sets $\mathcal{G}$ and $\mathcal{G}'$ and with $S' = S \sim D^n$. Given $L \in \mathbb{N}_+$, we define the sets of functions $\mathcal{G}$ and $\mathcal{G}'$ by

$$\mathcal{G} := \left\{ g : (x, y) \mapsto \max_{\varepsilon_l \in \varepsilon} H(f'(x + \varepsilon_l), y) \,\middle|\, \varepsilon = [\varepsilon_1, \dots, \varepsilon_L] \in \mathcal{B}_p(\delta)^L \right\},$$

$$\text{and} \quad \mathcal{G}' := \left\{ g : (x, y) \mapsto \max_{\varepsilon_l \in \varepsilon} H(f(x + \varepsilon_l), y) \,\middle|\, f \in \mathcal{F}, \ \varepsilon = [\varepsilon_1, \dots, \varepsilon_L] \in \mathcal{B}_p(\delta)^L \right\}.$$

Hence, by applying Lemma 2, we have with probability at least $1 - \lambda$ over $S \sim D^n$

$$\forall f \in \mathcal{F}, \varepsilon \in \mathcal{B}_p(\delta)^L, \varepsilon' \in \mathcal{B}_p(\delta)^L, \ \left| R_D^{f'}(\varepsilon') - R_S^f(\varepsilon) \right| \leq \sup_{\varepsilon' \in \mathcal{B}_p(\delta)^L} \left| R_S^{f'}(\varepsilon') - R_S^f(\varepsilon) \right| + \sqrt{\frac{\ln \frac{2}{\delta}}{2n}}.$$

Finally, by setting $\varepsilon' = \varepsilon$, we have the desired result. $\qquad \square$

## D    EXPERIMENTAL SETTINGS

In this section, we further detail how the numerical experiments were conducted.

### D.1    MNIST EXPERIMENTS

**Data splitting and pre-processing.** The 60K samples of the training set undergo random affine transformations keeping the center invariant. To this effect, we use random rotations between $[11.25, +11.25]$ degrees and a random scaling selected in $[-0.825, +0.825]$. These deformed samples are used to learn $f$ while we randomly pick 500 original un-deformed images from the training dataset to learn the (generalized) universal attacks. The 10K images of the test set are used to evaluate the performance of the attacks. All images are flatten into 784 dimensional rescaled vectors so that the pixel intensity lies within $[0, 1]$.

**Model to attack.** We consider a differentiable model satisfying the KŁ property assumed in Theorem 4 (see Remark 1). To this effect, we resort to the simple multi-layer perceptron from (Cireşan et al., 2010) which manages to achieve under $1\%$ test accuracy. It is made of scaled hyperbolic tangent activation functions as well of an input layer, 8 hidden layers and an output linear layer of sizes $784 \times 1000$, $1000 \times 1000$ and $1000 \times 10$, respectively. The network is trained using a stochastic gradient descent with batch size 100 with a learning rate linearly decreasing from $10^{-3}$ to $10^{-6}$ over $10^3$ epochs.

### D.2    CIFAR10 EXPERIMENTS

**Data splitting and pre-processing.** If not mentioned otherwise, we split CIFAR10 test set into 2K images for learning (generalized) universal perturbations and 8K independent images for evaluating the attacks.

**Model to attack.** We consider the trained ResNet18 model from (Phan, 2021) augmented with an input normalizing layer of channel-wise means $(0.4914, 0.4822, 0.4465)$ and channel-wise standard deviations $(0.2471, 0.2435, 0.2616)$.

### D.3    IMAGENET EXPERIMENTS

**Data splitting and pre-processing.** We resort to the popular ILSVRC2012 validation subset of the ImageNet dataset. The 50k images are randomly split into two halves. The first half is used to learn the universal perturbations while the second half is regarded as test set to evaluate the attacks. All images are resized into $256 \times 256$ followed by a cropping of size $224 \times 224$ around the center and a rescaling of the pixels intensity into $[0, 1]$. Results are averaged over 5 splits.

**Model to attack.** We analyze a pretrained ResNet18 model from the Torchvision library augmented with a normalizing layer of mean $(0.485, 0.456, 0.406)$ and standard deviation $(0.229, 0.224, 0.225)$ achieving a test accuracy of $69.76\%$.

$L$**-UAP solver.** Contrary to the previous experiments, we consider the ProxSAGA solver of Algorithm 2 in order to learn the $L$-UAP perturbations. The step-size and batch-size are set to $\gamma = 0.05$ and $b = 1$, respectively.

## E    ADDITIONAL RESULTS

In the next sections, we provide complementary results on both MNIST and CIFAR10 datasets.

### E.1    MNIST EXPERIMENTS

We analyze the training behavior of $L$-UAP attacks with $L \in \{1, 3, 5, 10\}$ universal perturbations learned with the Algorithm 1. The experiment is repeated over 5 independent seeds and the averaged training loss is reported in Figure 6. Independently of $L$, it shows the well-behaved increasing behavior of the loss along the number of epochs. In addition, it supports the fact that

having more universal perturbations does permit to achieve higher dissimilarity hence higher loss values. This is seconded by the mean test fooling rate reported for each of the $L$-UAP attacks since we observe an increased fooling rate as $L$ grows. On a side note, on this simple dataset, it is difficult to fool the studied network $f$, hence justifying the small fooling rates depicted in Figure 6.

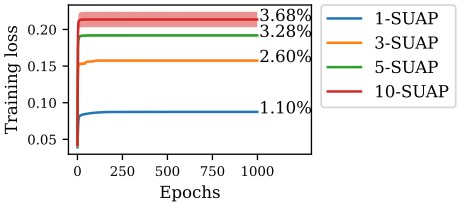

Figure 6: **Training behavior of $\ell_\infty$-based $L$-UAP attacks on MNIST**. The averaged training loss is reported for 1, 3, 5 and 10 universal perturbations along with the associated test fooling rate.

### E.2 CIFAR10 EXPERIMENTS

**Comparison between $\ell_\infty$ and $\ell_2$ attacks.** We report in Table 3 the performance comparison of both $\ell_\infty$ and $\ell_2$-attacks with a maximum allowable budget of $\delta = 8/255$ and $\delta = 0.5$, respectively.

**Comparison of UAP solvers for $L = 1$ perturbation.** We compared our proposed deterministic solver from Algorithm 1 with the stochastic solvers advocated in (Shafahi et al., 2020)), namely SGD and Adam with a batch size of 128. All three solvers are initialized identically and are run until convergence is reached. Learning rates are cross-validated in a fine logarithmic grid between $10^{-1.5}$ and $10^3$. To eliminate any potential influence of our different loss function on the observed performance improvement, we thoroughly examined all three loss functions $H$. More precisely, we have considered the cross-entropy (*ce*), the bounded cross entropy (*bce*) Dziugaite & Roy (2018) and the capped cross entropy (*max-ce*) introduced by the authors of the UAP-PGD attack (Shafahi et al., 2020)). The test fooling rate, averaged over multiple data splits, are reported in Figure 7. Interestingly, the choice of the loss do not impact significantly the results. In addition, both SGD and Adam yield similar performance, while our deterministic solver substantially improves the quality of the learned attack.

Table 3: **Performance of attacks on a ResNet18 trained on CIFAR-10.** Bold fonts highlight the best fooling rate in universal (top), sproposed (middle) and specific (bottom) attacks.

| Attack | $\ell_\infty$-fooling rate (%) | $\ell_2$-fooling rate (%) |
|---|---|---|
| UAP-PGD (Shafahi et al., 2020) | 12.53 ($\pm$ 0.60) | 2.67 ($\pm$ 0.21) |
| FAST-UAP (Dai & Shu, 2021) | 11.16 ($\pm$ 1.03) | 2.53 ($\pm$ 0.19) |
| CW-UAP (Benz et al., 2021) | **13.85 ($\pm$ 0.18)** | **2.77 ($\pm$ 0.09)** |
| 1-UAP | 36.83 ($\pm$ 0.93) | 3.43 ($\pm$ 0.26) |
| 3-UAP | 54.03 ($\pm$ 0.54) | 4.93 ($\pm$ 0.50) |
| 5-UAP | **55.56 ($\pm$ 0.57)** | **7.09 ($\pm$ 1.22)** |
| FGSM (Goodfellow et al., 2015) | 53.82 ($\pm$ 0.00) | N/A |
| MI-FGSM (Dong et al., 2018) | 80.76 ($\pm$ 0.00) | N/A |
| PGD (Madry et al., 2018) | **93.61 ($\pm$ 0.06)** | 89.23 ($\pm$ 0.02) |
| AutoAttack (Croce & Hein, 2020) | 93.07 ($\pm$ 0.00) | **92.41 ($\pm$ 0.01)** |

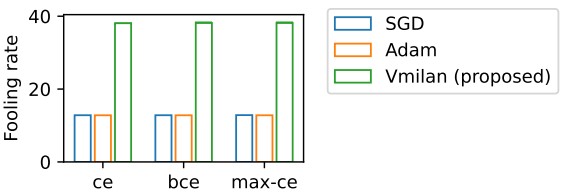

Figure 7: **Comparison of solvers for** $L = 1$**.** The averaged test fooling rate is reported when learning a single perturbation with multiple solvers and different type of losses $H$. We have considered the cross entropy (ce), its bounded variant (bce) and its clamped version (max-bce).

