# OpenReview forum: "A Theoretically Grounded Extension of Universal Attacks from the Attacker's Viewpoint"
_ICLR.cc/2024/Conference — ICLR 2024 Conference Withdrawn Submission_

### Official Review · Reviewer_y5za · 2023-10-29

**Soundness:** 2 fair
**Presentation:** 2 fair
**Contribution:** 2 fair
**Rating:** 5
**Confidence:** 4

**Summary:**

The paper views the universal attack problem from a perspective of generalization bound with Radamecher complexity analysis and accordingly proposes an effective optimization procedure to learn the universally effective adversarial perturbations.

**Strengths:**

1) The paper clarifies well, especially in the preliminary and related work part.

2) Viewing the universal attack from a generalization analysis perspective is novel to me.

**Weaknesses:**

1) The effectiveness of the L-UAP algorithm is not adequately explained. It basically designs a new step update procedure (inner loop of algorithm 1). How is that important? Is there any analytical or theoretical grounding?

2) The proof of Theorem 4 is missing. Although one paper is cited, and the theorem may be built from it, I do not think the paper considers Algorithm 1 proposed by you, so it was important to show the proof of it.

3) From the results in Table 1, why are multiple perturbations beneficial for source model attack but not that good for transfer attack?

4) Problem 1 and Algorithm 1 only do optimization with respect to one model, which is similar to a single model attack. What is the special component here for universal attack?

5) In proposition 1, the paper defines Radamecher complexity of a set of perturbations $\mathcal{B}_p(\delta)$, which is weird to me since the complexity measure is defined for a function class.

6) In Theorem 2, consider the formula inside the probability operator, parameter $L$ does not appear in LHS. What the paper bounds here is exactly the same as that in Theorem 1.

7) Is it possible to consider the complexity measure (Rademacher complexity) as a regularization term to improve the transferability of universal attacks? If it is feasible, the idea looks more interesting.

8) Presentation: I do not even find the definitions of $\rho$ and $h$ in algorithm 1 in the main text, which is very confusing to readers.

**Questions:**

Please refer to the weakness part.

---

### Official Review · Reviewer_sp9Z · 2023-10-29

**Soundness:** 3 good
**Presentation:** 2 fair
**Contribution:** 2 fair
**Rating:** 3
**Confidence:** 4

**Summary:**

The paper proposes a new method for generating universal adversarial perturbations by optimizing multiple perturbations simultaneously and selecting the most optimal one.

**Strengths:**

* The paper is easy to follow.
* The issue of universal adversarial perturbation is quite intriguing.

**Weaknesses:**

* The theoretical contribution is incremental and offers limited guidance for algorithm design. The primary theoretical input is the application of Rademacher complexity to the generalization of adversarial perturbations, which is somewhat limited. Furthermore, this theoretical insight does not significantly influence the algorithm's design.

* The experiments on adversarial transferability are not comprehensive enough. (1) Tests were conducted only on CIFAR-10, whereas most experiments concerning adversarial transferability are performed on ImageNet. (2) The choice of the target model is not reasonable; since the source model is already a ResNet, the target model should incorporate structures other than ResNet.

* The motivation for Equation (9) is unclear. Regarding Equation (9), it is not explained why a set of perturbations is used. Also, the definition of problem 1 suggests there should be constraints on L; otherwise, when L equals n, Equation (9) essentially degenerates into Equation (1). Moreover, there is a lack of discussion in the experimental section regarding the appropriate size of the parameter L.

* Comparative experiments are not entirely fair. The authors' method optimizes several perturbations simultaneously, but the baseline methods compared are all optimizing a single perturbation. This comparison's fairness needs further discussion.

**Questions:**

* The MI-FGSM is designed to improve adversarial transferability, but why does MI-FGSM not outperform PGD in Table 1?

---

### Official Review · Reviewer_fSqy · 2023-10-31

**Soundness:** 2 fair
**Presentation:** 4 excellent
**Contribution:** 3 good
**Rating:** 3
**Confidence:** 3

**Summary:**

The authors consider a generalization of universal attacks in which, for any input, the attacker can choose the optimal attack from a fixed set of perturbations. Under this model, the authors prove generalization bounds on the fooling risk. Empirical results demonstrating the attack are then presented.

**Strengths:**

The way universal attacks are generalized leads to a novel adversarial attacking framework. The ideas are presented clearly, and contextualized well.

**Weaknesses:**

It is unclear how good the generalization bounds are. As a result it is difficult to evaluate the theoretical results. (More details in the Questions section)

**Questions:**

On page 4, the claim that Equation 6 holds for any model $f$ we want to attack that is not necessarily the one we used to learn $\varepsilon$ appears to be false.

The generalization bounds (Proposition 1 and Theorem 2) are given in terms of Rademacher complexity. What is the Rademacher complexity? Does it converge to 0? If so, how quickly? There is no indication of how large this term is. Are the generalization bounds vacuous?

On page 7, the estimation of the Rademacher complexity gives an approximate lower bound of the Rademacher complexity, but the Rademacher complexity is used as an upper bound. As a result I am not sure how useful this estimate is.

The bound given in Proposition 3 also seems quite bad. In particular, the term $sup_{\varepsilon'}|R_S^{f'}(\varepsilon')-R_S^f(\varepsilon)|$. It is claimed that "this term is small if we cannot find a set of perturbations $\varepsilon'$ that differs too much from the perturbations $\varepsilon$ for $f'$." But isn't it almost always possible to find such a set? As an example, you could set $\varepsilon'$ so that all of its elements have almost zero norm, so there is essentially zero perturbation.

---

### Official Review · Reviewer_P53G · 2023-10-31

**Soundness:** 2 fair
**Presentation:** 2 fair
**Contribution:** 2 fair
**Rating:** 5
**Confidence:** 4

**Summary:**

This paper introduces an extension to the Universal Adversarial Perturbation (UAP) by developing a set of perturbations, termed L-UAP, which are jointly learned to optimize the likelihood of successfully attacking deep neural network models.

**Strengths:**

Theoretically, the research broadens the scope of Rademacher complexity from a model learning perspective to encompass perturbation learning, aiming to gauge the generalization capability of universal attacks.

Experimentally, the paper provides comprehensive experiments. The trends of attack rates w.r.t the number of perturbation L provides information of readers to understand the ability of UAP.

**Weaknesses:**

The primary limitation of this research is the weak linkage between the theoretical framework and experimental results, even though it is a challenge commonly observed in uniform convergence analyses.

1) In both Theorem 1 and Theorem 2, the bounds are set for all perturbations within $B_p(\delta)$, rather than specifically for learned UAP attacks.

2) When contrasting Theorem 1 with Theorem 2, it becomes evident that the generalization abilities of UAP and L-UAP are evaluated by $R(B_p(\delta))$ and $R(B_p(\delta)^L)$ respectively. Considering that $B_p(\delta)^L$ represents a more extensive class, its Rademacher complexity would naturally be larger. As a result, the theorem might suggest that L-UAP's generalization is inferior to UAP, which contradicts the paper's primary objective. This is similar to a general criticism for previous rademacher complexity analysis: Rademacher complexity of large model is very large, yet large model generalize better.

Therefore, without a deeper analysis of $R(B_p(\delta))$ and $R(B_p(\delta)^L)$, the analysis is not informative. Given this, the assertion in Section 2 — "motivated by Theorem 2, we propose to maximize problem 1” — doesn't come across as compelling.

3) Theorem 4's convergence results under the Lipschitz and KL conditions appear somewhat lacking in depth and specificity.

Given the weak theoretical analysis, increasing the number of perturbation candidates is not very novel.

**Questions:**

See weakness

---

### Author Response · Authors · 2023-11-21

Dear Reviewers,

Your time and considerate feedback on our submission are deeply valued. Every suggestion and concerns you provided has been meticulously examined. We have taken a comprehensive approach in addressing your comments and integrating them into our revised manuscript.

However, we believe that the required clarifications about both our theoretical bound and the applicability of the algorithmic solution cannot be integrated to meet the page limit.

We have made the decision to withdraw our paper.

Sincerely